# Lipid profiles and risk of major adverse cardiovascular events in CKD and diabetes: A nationwide population-based study

Yeonhee Lee[1,2], Sehoon Park[3,4], Soojin Lee[1,2], Yaerim Kim[5], Min Woo Kang[1,2], Semin Cho[1,2], Sanghyun Park[6], Kyungdo Han[6], Yong Chul Kim[1,7], Seoung Seok Han[1,7], Hajeong Lee[1,7], Jung Pyo Lee[2,7,8], Kwon Wook Joo[1,2,7], Chun Soo Lim[2,7,8], Yon Su Kim[3,4,7], Dong Ki Kim[3,4,7]*

1 Department of Internal Medicine, Seoul National University Hospital, Seoul, Korea, 2 Department of Internal Medicine, Seoul National University College of Medicine, Seoul, Korea, 3 Department of Biomedical Sciences, Seoul National University College of Medicine, Seoul, Korea, 4 Department of Internal Medicine, Armed Forces Capital Hospital, Gyeonggi-do, Korea, 5 Department of Internal Medicine, Keimyung University School of Medicine, Daegu, Korea, 6 Department of Medical Statistics, College of Medicine, Catholic University of Korea, Seoul, Korea, 7 Kidney Research Institute, Seoul National University College of Medicine, Seoul, Korea, 8 Department of Internal Medicine, Seoul National University Boramae Medical Center, Seoul, Korea

* dkkim73@gmail.com

**Data Availability Statement:** This study was performed using the Korean National Health Insurance Service (NHIS) and the Health Insurance Review and Assessment Service (HIRA) claims

## Abstract

The association of lipid parameters with cardiovascular outcomes and the impact of kidney function on this association have not been thoroughly evaluated in chronic kidney disease (CKD) patients with diabetes. We reviewed the National Health Insurance Database of Korea, containing the data of 10,505,818 subjects who received routine check-ups in 2009. We analyzed the association of lipid profile parameters with major adverse cardiovascular events (MACEs) risk and all-cause mortality in a nationally representative cohort of 51,757 lipid-lowering medication-naïve patients who had CKD and diabetes. Advanced CKD patients with eGFR <30 mL/min/1.73 m$^2$ (n = 10,775) had lower serum total cholesterol (TC), low-density lipoprotein cholesterol (LDL-c), and high-density lipoprotein cholesterol (HDL-c) but higher non-HDL-c levels and triglyceride (TG) to HDL-c ratios. There was a positive linear association between serum LDL-c and MACE risk in both early and advanced CKD patients (*P* <0.001 for trend), except for the category of LDL-c 30–49 mg/dL in extremely low LDL-c subgroup analyses. A U-shaped relationship was observed between serum LDL-c and all-cause mortality (the 4[th] and 8[th] octile groups; lowest hazard ratio [HR] 0.96, 95% confidence interval [CI] 0.87–1.05 and highest HR 1.14, 95% CI 1.04–1.26, respectively). A similar pattern remained in both early and advanced CKD patients. The TG/HDL-c ratio categories showed a positive linear association for MACE risk in early CKD (*P* <0.001 for trend), but this correlation disappeared in advanced CKD patients. There was no correlation between the serum TG/HDL-c ratio and all-cause mortality in the study patients. The LDL-c level predicted the risk for MACEs and all-cause mortality in both early and advanced CKD patients with diabetes, although the patterns of the association differed from each other. However, the TG/HDL-c ratio categories could not predict the risk for either

databases. Access to the HIRA database is restricted, being permitted only after approval by the HIRA Deliberative Committee for studies that are conducted for the common good. Data are available through the Korean National Health Insurance Sharing Service. Researchers who wish to access the data can apply at (https://nhiss.nhis.or.kr/bd/ay/bdaya001iv.do) and request access to NHIS-2018-1-220.

**Funding:** This work was supported by a grant from the Korea Healthcare Technology R&D Project, the Ministry of Health and Welfare, Republic of Korea (HI17C0530).

**Competing interests:** The authors have declared that no competing interests exist.

MACEs or all-cause mortality in advanced CKD patients with diabetes, except that the TG/HDL-c ratio predicted MACE risk in early CKD patients with diabetes.

## Introduction

Cardiovascular disease (CVD) is the leading cause of mortality in chronic kidney disease (CKD) patients [1, 2]. Therefore, the early determination and management of the risk factors for CVD in CKD patients play an essential role in treatment strategies to decrease cardiovascular mortality and morbidity in CKD patients. It is well known that even mildly reduced kidney function represents a major risk factor for atherosclerotic CVD [3, 4]. Additionally, it was previously documented that one of the most important pathophysiological mechanisms of CVD in patients with CKD is the widespread and possibly accelerated formation of atherosclerotic plaques due to dyslipidemia [5, 6].

Patients with kidney disease form a heterogeneous population with various etiologies of kidney damage, levels of kidney function and proteinuria, and comorbidities, all of which can affect the levels and properties of circulating lipids [7]. The characteristic lipid pattern in these patients shows a different profile from the dyslipidemia of the general population, consisting of hypertriglyceridemia, low levels of high-density lipoprotein cholesterol (HDL-c) and variable levels of low-density lipoprotein cholesterol (LDL-c) and total cholesterol (TC) [8–10]. In particular, diabetes is the main risk factor for end-stage renal disease (ESRD) and the most advanced stage of CKD. Underlying dysglycemia further accelerates the kidney damage induced by dyslipidemia [11]. Thus, dyslipidemia and kidney diseases act synergistically to worsen the clinical condition and increase the risk of kidney or cardiovascular consequences among diabetic patients. Recently, the association of lipid parameters with CVD risk in CKD patients has become an area of great interest. Several studies have shown an inverse relationship between LDL-c levels and all-cause mortality in patients with ESRD [12]. Moreover, low levels of HDL-c are common among patients with CKD and ESRD, but they do not seem to be associated with increased cardiovascular risk [13–15]. Furthermore, contrary to the general population, an elevated triglyceride to HDL-c (TG/HDL-c) ratio was associated with better cardiovascular and overall survival in patients on hemodialysis [16].

Although lipid metabolism disorders are more frequent in these patients, the association of lipid parameters with major adverse cardiovascular events (MACEs) and mortality has not been thoroughly evaluated in CKD patients with diabetes. The main purpose of our study was to identify possible associations between lipid profile parameters and MACEs and mortality in CKD patients with diabetes and to evaluate the impact of kidney function on the associations.

## Materials and methods

### Study population and data source

This study was performed using the Korean National Health Insurance Service (NHIS) and the Health Insurance Review and Assessment Service (HIRA) claims databases. Access to the HIRA database is restricted, being permitted only after approval by the HIRA Deliberative Committee for studies that are conducted for the common good. Data are available through the Korean National Health Insurance Sharing Service. Researchers who wish to access the data can apply at (https://nhiss.nhis.or.kr/bd/ay/bdaya001iv.do) and request access to NHIS-2018-1-220. The NHIS, a mandatory form of social insurance, covers about 97% of the Korean population. The NHIS offers general health and cancer-screening programs. All insured or

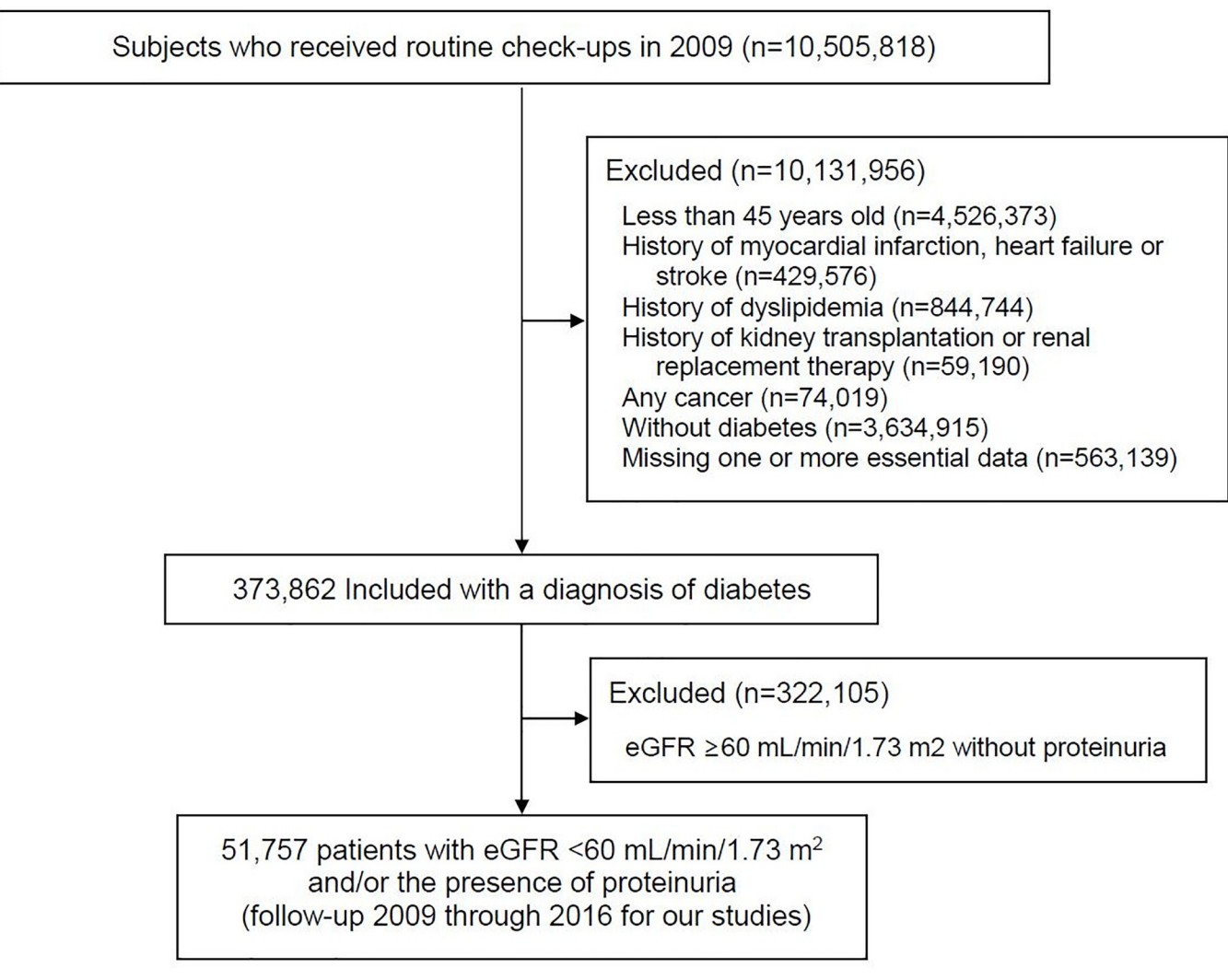

**Fig 1. Flow chart of a cohort of 51,757 patients in the final analyses.** eGFR, estimated glomerular filtration rate.

self-employed persons over the age of 40 are entitled to free health checkups. The HIRA database contains health care utilization information, including demographic characteristics, diagnoses (with 10th International Classification of Diseases [ICD-10] codes), medical procedures, prescription records, and direct medical costs. We were provided access to the data for the years 2009 through 2016 for our studies. The cohort contains the data of 10,505,818 subjects who received routine check-ups in 2009 and includes demographic data, eligibility status, income levels, claims, and death records through the end of 2016. The inclusion and exclusion criteria for participants are shown in the flow chart in Fig 1. Of 10,505,818 participants, we excluded 563,139 participants without essential data, including age; sex; body mass index (BMI); waist circumference; information on smoking and alcohol consumption; levels of serum creatinine, fasting glucose, total cholesterol (TC), LDL-c, HDL-c, and TGs; and the use of medications for hypertension, diabetes mellitus, and dyslipidemia. A total of 9,568,817 subjects with less than 45 years of age; a previous history of myocardial infarction, heart failure, stroke, dyslipidemia, kidney transplantation, renal replacement therapy or any cancer; and without diabetes were also excluded from the analysis. We then selected 373,862 patients who had received a diagnosis of

diabetes (ICD-10 codes E11, E13, or E14) as the principal diagnosis on at least 2 occasions within the last 3 years. Finally, with data for eGFR and urinary protein levels, 51,757 participants (27,666 men and 24,091 women) were analyzed in the present study. The insurance eligibility database was linked to data from the National Database of Statistics Korea by using the Korean resident registration number. The study was approved by the Seoul National University Hospital Institutional Review Board (E-1801-105-917) and was conducted in accordance with the tenets of the Declaration of Helsinki. The informed consents were waived by the IRB, because this study was a retrospective study and the NHIS database was anonymized for research purposes.

## Definitions

The estimated glomerular filtration rate (eGFR) was calculated using the MDRD equation: eGFR (mL/min/1.73 m$^2$) = 194 × serum creatinine (mg/dL)$^{-1.094}$ × age(years) $^{-0.287}$ × 0.739 (for women) [17]. CKD was defined as eGFR < 60 mL/min/1.73 m$^2$ and/or the presence of proteinuria. Early CKD was defined as a confirmed (two consecutive measurements ≥3 months apart) eGFR ≥30 and <60 mL/min/1.73 m$^2$, and advanced CKD was defined as eGFR <30 mL/min/1.73 m$^2$. Urinary protein excretion was examined by dipstick testing and categorized into 5 degrees; -, ±, 1+, 2+, and 3+. Proteinuria was defined as urinary protein ≥1+. Fasting glucose, TC, LDL-c, HDL-c, and TGs were measured using standard laboratory methods. LDL-c was calculated according to the Friedewald formula, and non-HDL-c was calculated by subtracting HDL-c level (mg/dL) from TC level (mg/dL). TG/HDL-c ratio was calculated as TG level (mg/dL) divided by HDL-c level (mg/dL). Non-HDL/HDL-c ratio was calculated as non-HDL-c level (mg/dL) divided by HDL-c level (mg/dL). Central obesity was defined as waist circumference (WC) ≥90 cm for men and ≥85 cm for women [18].

The exposures of interest were serum LDL-c and the TG/HDL-c ratio. Given a possible nonlinear relationship with CVD risk and mortality rates, the parameters were treated as categorical variables and divided into octiles. The reference LDL-c and TG/HDL ratio category for all analyses was the fifth octile, with the exception of LDL-c in the advanced CKD group. This category was chosen as the reference because the median ratio in this study was similar to the normal range that was used in previous studies and derived from the Adult Treatment Panel recommendations (on the basis of optimal LDL-c <110 mg/dL, normal fasting TGs <150 mg/dL and HDL-c >40 mg/dL) [19]. Because LDL-c is normalized or even reduced in more advanced stages of CKD [20], the reference LDL-c level category for patients with advanced CKD was the third octile, which has the lowest hazard ratio (HR).

The primary outcome was the occurrence of MACEs, defined as a composite of nonfatal myocardial infarction, heart failure, and stroke. Nonfatal myocardial infarction was defined as a hospitalization with the ICD-10 code I21 as the primary or secondary diagnosis and ICD-10 revascularization procedure codes. Heart failure and stroke were defined by discharge diagnoses (ICD-10 code: I59 and I63, respectively) after hospitalization. The secondary outcome was all-cause mortality.

## Statistical analyses

Baseline characteristics were compared using the Wilcoxon rank-sum test for continuous variables and the chi-square test for categorical variables. We used Cox proportional hazard regression models separately to analyze the associations of baseline and time-varying lipid profile parameters with cardiovascular disease and mortality. In the baseline models, lipid profile parameters and covariates were calculated at baseline, and their association with cardiovascular disease and mortality was analyzed. In time-varying models, lipid profile parameters and covariates were determined and reassessed for each patient-year over the entire period of

follow-up to evaluate short-term associations between lipid profile parameters and cardiovascular disease risk, assuming that lipid profile parameters remained unchanged during the time interval before the next measurement. For each analysis, unadjusted and multivariate adjustment, which adjusted for baseline characteristics of age, sex, body mass index (BMI), smoking, systolic blood pressure, diastolic blood pressure, eGFR, urinary protein, serum hemoglobin, and glucose level, were performed. All mortality associations are expressed as hazard ratios and 95% confidence intervals. Given that lipid metabolism may differ according to renal function, we further assessed the association of lipid profile parameters with CVD risk and all-cause mortality divided by an early or advanced stage of CKD. All analyses were implemented using the SAS 9.4 program (SAS Institute).

## Results

### Study population

The study population comprised 51,757 patients with CKD and diabetes who had no history of major cardiovascular events. Table 1 shows selected baseline characteristics of the study patients. Among lipid-lowering medication-naïve study patients extracted from the National Health Insurance Services Health Screening cohort, 10,775 (20.8%) had an advanced stage of CKD (eGFR <30 mL/min/1.73 m$^2$). The mean age of patients was 63.4±10.5 years in the early CKD group and 64.3±10.9 years in the advanced CKD group. Patients with advanced CKD, when compared with patients in the early CKD group, tended to be older women whose waist circumference was smaller. Approximately 18% of the study participants were current smokers. The mean serum creatinine level was 1.19±0.17 mg/dL in the early CKD group and 3.97 ±3.25 mg/dL in the advanced CKD group. Advanced CKD patients also had lower serum TC, LDL-c, and HDL-c but higher non-HDL-c levels and TG/HDL-c ratios (Table 1).

### Cardiovascular disease risk and all-cause mortality: LDL-c

During the median follow-up time of 7.3 years, 6,555 (12.7%) CVD events and 7,289 (14.1%) all-cause deaths occurred. From the lowest (first) to highest (8th) octiles of baseline LDL-c, MACE and all-cause mortality rates were 19.3, 18.2, 18.4, 19.0, 19.0, 18.8, 18.9, and 21.5 and 26.5, 23.6, 21.3, 18.6, 19.1, 17.2, 17.3, and 18.0 per 1000 patient-years, respectively. The HRs for the categories of LDL-c serum levels calculated using multivariate Cox regression analysis, as well as adjusted for age, sex, BMI, smoking, systolic blood pressure, diastolic blood pressure, eGFR, urinary protein, serum hemoglobin, and glucose level, are shown in Table 2. In the baseline models, there was a positive linear association between serum LDL-c and MACE risk from the lowest to highest octiles in patients with CKD and diabetes. The lower LDL-c category had a lower HR for MACE risk in CKD patients with diabetes. Lower LDL-c lost its beneficial effect on the risk of MACEs and all-cause mortality in time-varying models.

We next performed subgroup analyses to determine the effect of LDL-c levels on MACE risk and mortality in patients stratified into two groups by eGFR. The reference LDL-c level category for patients with advanced CKD was the third octile, which had the lowest HR. In the baseline models, a positive linear association was observed between serum LDL-c and MACE risk from the lowest to highest octiles in patients with both early (Fig 2A) and advanced CKD (Fig 2B). Interestingly, this association was abolished after additional adjustment in time-varying models.

The highest octile group of baseline LDL-c had the highest all-cause mortality in both groups. However, there was no linear correlation between serum LDL-c and all-cause mortality, with significantly higher mortality in octiles 1–2 and 7–8 (a "U" shaped curve) (Fig 2C). Notably, this U-shaped relationship was more prominent in patients with advanced CKD (Fig 2D) and remained largely unchanged in the time-varying models.

**Table 1. Baseline characteristics of study patients stratified by stages of chronic kidney disease.**

| | Total (n = 51,757) | Diabetes with Early CKD (n = 40,983) | Diabetes with Advanced CKD (n = 10,775) |
|---|---|---|---|
| Age (yr) | 63.6 ± 10.6 | 63.4 ± 10.5 | 64.3 ± 10.9 |
| Male sex (n (%)) | 27,666 (53.5) | 22,255 (54.3) | 5411 (50.2) |
| Body mass index (kg/m²) | | | |
| <18.5 (n (%)) | 944 (1.8) | 718 (1.8) | 226 (2.1) |
| 18.5–23 (n (%)) | 13,975 (27.0) | 10,699 (26.1) | 3276 (30.4) |
| 23–25 (n (%)) | 13,441 (26.0) | 10,663 (26.0) | 2778 (25.8) |
| 25–30 (n (%)) | 20,382 (39.4) | 16,506 (40.3) | 3876 (36.0) |
| ≥30 (n (%)) | 3015 (5.8) | 2396 (5.9) | 619 (5.7) |
| Waist circumference (cm) | 85.3 ± 8.4 | 85.4 ± 8.4 | 84.9 ± 8.5 |
| Central obesity[a] (n (%)) | 25,972 (50.2) | 20,677 (50.5) | 5295 (49.1) |
| Smoking (n (%)) | | | |
| Never | 33,198 (64.1) | 26,246 (64.0) | 6952 (64.5) |
| Ex-smoker | 8890 (17.2) | 7000 (17.1) | 1890 (17.5) |
| Current smoker | 9669 (18.7) | 7736 (18.9) | 1933 (17.9) |
| Alcohol (n (%)) | | | |
| Never | 33,998 (65.7) | 26,742 (65.3) | 7256 (67.3) |
| Moderate | 14,332 (27.7) | 11,417 (27.9) | 2915 (27.1) |
| Heavy | 3427 (6.6) | 2823 (6.9) | 604 (5.6) |
| Low Income level[b] (n (%)) | 12,118 (23.4) | 9944 (24.3) | 2174 (20.2) |
| Systolic blood pressure (mmHg) | 130.5 ± 16.2 | 130.2 ± 16.1 | 131.4 ± 16.6 |
| Diastolic blood pressure (mmHg) | 79.1 ± 10.2 | 79.2 ± 10.1 | 78.8 ± 10.3 |
| eGFR (mL/min/1.73 m²) | 48.7 ± 13.9 | 54.5 ± 3.9 | 26.6 ± 16.2 |
| Serum creatinine (mg/dL) | 1.8 ± 1.9 | 1.2 ± 0.2 | 3.9 ± 3.2 |
| Fasting glucose (mg/dL) | 144.5 ± 45.7 | 145.0 ± 45.4 | 142.6 ± 46.8 |
| Lipid parameters | | | |
| Total cholesterol (mg/dL) | 203.1 ± 39.6 | 203.8 ± 39.4 | 200.5 ± 40.4 |
| LDL-c (mg/dL) | 118.0 ± 36.1 | 118.3 ± 36.1 | 116.8 ± 36.0 |
| TGs (mg/dL) | 149.9 ± 0.7 | 149.9 ± 0.8 | 150.0 ± 1.5 |
| HDL-c (mg/dL) | 50.3 ± 13.2 | 50.6 ± 13.2 | 49.0 ± 13.2 |
| Non-HDL-c (mg/dL) | 152.8 ± 38.4 | 153.2 ± 38.2 | 151.6 ± 39.1 |
| TG/HDL-c ratio | 3.8 ± 3.1 | 3.8 ± 3.0 | 4.0 ± 3.2 |
| Non-HDL-c/HDL-c ratio | 3.3 ± 1.3 | 3.3 ± 1.3 | 3.3 ± 1.3 |

Data are presented as proportions or means ± standard deviations.

[a] Central obesity was defined as waist circumference (WC) ≥90 cm for men and ≥85 cm for women.

[b] Low income was defined as a total income <20th percentile for the nation.

CKD, chronic kidney disease; eGFR, estimated glomerular filtration rate; LDL-c, low-density lipoprotein cholesterol; TG, triglycerides; HDL-c, high-density lipoprotein cholesterol.

Next, we conducted subgroup analyses to detect the effect of LDL-c levels on MACE risk and mortality in patients with extremely low LDL-c (<79 mg/dL), by dividing into three subgroups of LDL-c 50–78 mg/dL, 30–49 mg/dL and <30 mg/dL. When compared with patients with LDL-c 50–78 mg/dL, the relative hazard of MACEs and all-cause mortality in the category of LDL-c 30–49 mg/dL was higher in both baseline and time-varying models (highest HR 1.32, 95% CI 1.13–1.55). Notably, this trend was more prominent in patients with advanced CKD (HR 1.83, 95% CI 1.36–2.45 and HR 1.80, 95% CI 1.34–2.42 in baseline and time-varying models, respectively) (S1 and S2 Tables).

### Cardiovascular disease risk and all-cause mortality: TG/HDL-c ratio

From the lowest (first) to highest (8th) octiles of baseline TG/HDL-c ratio, MACE and all-cause mortality rates were 14.4, 16.3, 18.1, 19.3, 20.4, 20.7, 22.1, and 21.8, and 19.9, 20.2, 21.6, 21.8, 20.0, 20.0, 19.3, and 18.3 per 1000 patient-years, respectively. The HRs for the categories of TG/HDL-c ratios calculated using multivariate Cox regression analysis are shown in Table 3. A positive linear association was observed between the serum TG/HDL-c ratio and MACE risk from the lowest to highest octiles in CKD patients with diabetes in the baseline models. This pattern weakened after additional adjustment in the time-varying models. There was no correlation overall between the serum TG/HDL-c ratio and all-cause mortality, except in octiles 1–2 (TG/HDL-c ratio <1.95).

In subgroup analyses, the TG/HDL-c ratio categories showed a positive linear association for MACE risk in patients with early CKD (Fig 3A), but this correlation disappeared in the advanced CKD group (Fig 3B). There was no correlation between the serum TG/HDL-c ratio and all-cause mortality (Fig 3C), and the noncorrelation was more salient in advanced CKD patients (Fig 3D). Overall, similar trends were observed for both all-cause and MACE risk in adjusted baseline and time-varying models.

## Discussion

Reduced kidney function, as well as diabetes, has been identified to increase the risk of mortality [21], cardiovascular events, and hospitalization [3], mainly because of a high prevalence of

**Table 2. Association of serum LDL-c with MACEs and all-cause mortality, stratified by octile categories in patients with CKD and diabetes.**

| MACE | | | Baseline model | | Time-varying model | |
|---|---|---|---|---|---|---|
| Level | N | Event | HR (95% CI) | *P* value | HR (95% CI) | *P* value |
| <79 | 6564 | 821 | 0.871 (0.789–0.961) | 0.006 | 0.976 (0.883–1.077) | 0.62 |
| 79–94 | 6492 | 779 | 0.889 (0.806–0.982) | 0.02 | 1.023 (0.927–1.13) | 0.64 |
| 95–105 | 6096 | 745 | 0.917 (0.83–1.014) | 0.09 | 1.013 (0.916–1.119) | 0.80 |
| 106–116 | 6693 | 847 | 0.995 (0.903–1.096) | 0.91 | 1.056 (0.958–1.163) | 0.26 |
| 117–127 | 6540 | 825 | 1 (Ref.) | | 1 (Ref.) | |
| 128–140 | 6581 | 826 | 1.023 (0.928–1.127) | 0.65 | 0.964 (0.875–1.063) | 0.46 |
| 141–158 | 6439 | 812 | 1.088 (0.987–1.2) | 0.09 | 0.972 (0.881–1.072) | 0.56 |
| ≥159 | 6352 | 900 | 1.261 (1.145–1.388) | <0.001 | 1.028 (0.934–1.132) | 0.57 |
| **All-cause mortality** | | | | | | |
| <79 | 6564 | 1190 | 1.024 (0.935,1.122) | 0.61 | 1.102 (1.006,1.207) | 0.04 |
| 79–94 | 6492 | 1058 | 1.095 (1,1.2) | 0.05 | 1.187 (1.084,1.3) | <0.001 |
| 95–105 | 6096 | 907 | 1.011 (0.92,1.11) | 0.82 | 1.067 (0.971,1.172) | 0.17 |
| 106–116 | 6693 | 875 | 0.956 (0.869,1.051) | 0.35 | 0.985 (0.896,1.084) | 0.76 |
| 117–127 | 6540 | 877 | 1 (Ref.) | | 1 (Ref.) | |
| 128–140 | 6581 | 796 | 0.959 (0.87,1.057) | 0.39 | 0.931 (0.844,1.025) | 0.14 |
| 141–158 | 6439 | 784 | 1.034 (0.938,1.141) | 0.49 | 0.962 (0.872,1.061) | 0.43 |
| ≥159 | 6352 | 802 | 1.141 (1.035,1.258) | 0.008 | 1.01 (0.916,1.114) | 0.84 |

Multivariate adjustment, which adjusted for baseline characteristics of age, sex, BMI, smoking, systolic blood pressure, diastolic blood pressure, eGFR, urinary protein, serum hemoglobin and glucose level.

In the baseline models, the TG/HDL-c ratio and covariates were determined at baseline, and their association with mortality was estimated. In the time-varying models, lipid profile parameters and covariates were calculated and updated each year over the entire follow-up period to assess short-term associations between lipid profile parameters and MACE risk, assuming that lipid profile parameters remained unchanged during the time interval before the next measurement.

TG/HDL-c, triglyceride/high-density lipoprotein cholesterol ratio; LDL-c, low-density lipoprotein cholesterol; MACE, major adverse cardiovascular event; CKD, chronic kidney disease; HR, hazard ratio; CI, confidence interval.

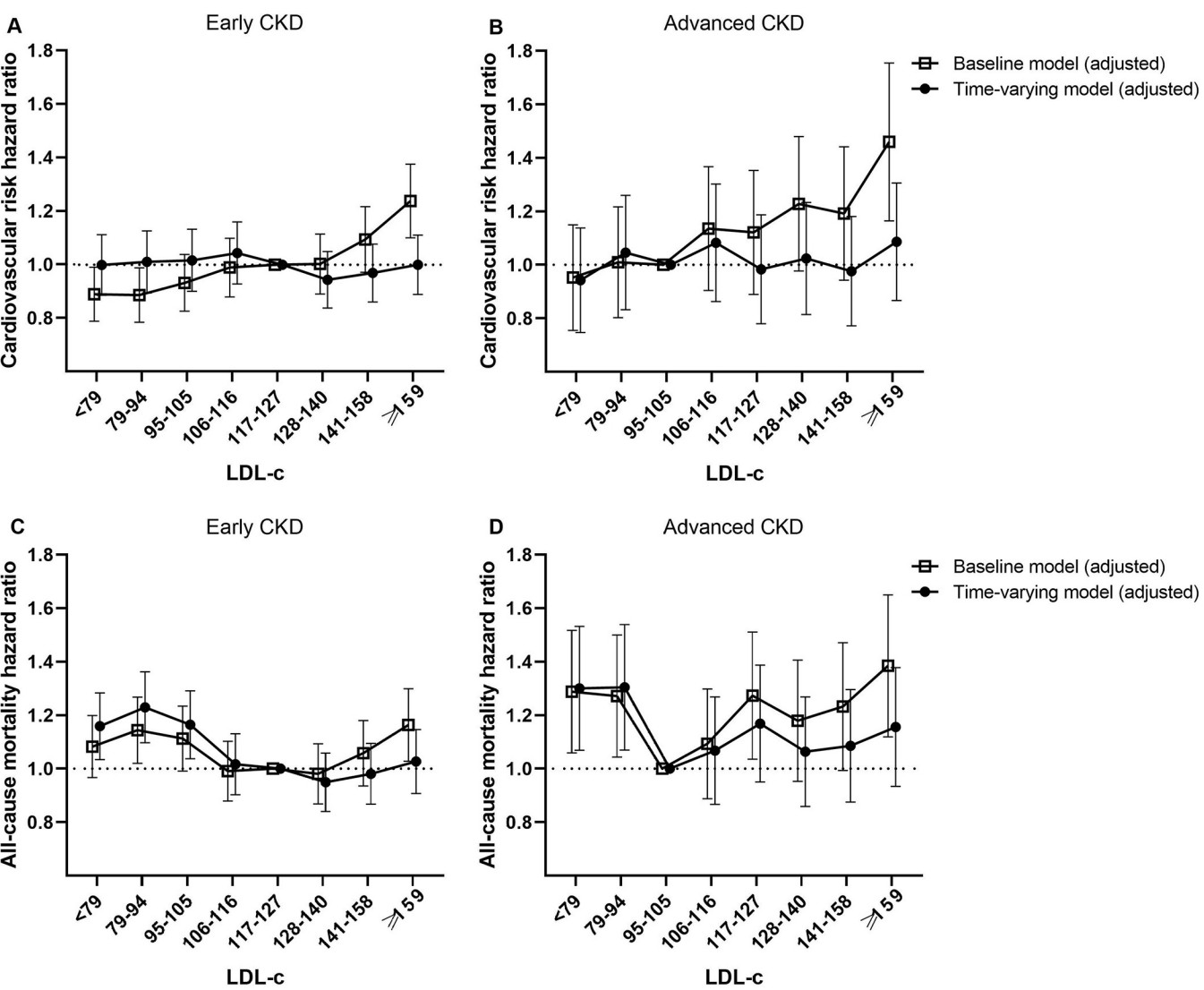

**Fig 2.** Multivariate-adjusted MACE (A and B) and all-cause mortality (C and D) hazard ratios by serum LDL-c level in early CKD and advanced CKD patients.

cardiovascular risk factors, including dyslipidemia [22]. CKD patients experience a secondary form of dyslipidemia that mimics the atherogenic dyslipidemia of insulin-resistant patients [23]. This dyslipidemia is characterized by an increase in serum TG with elevated very-low-density lipoprotein cholesterol, small dense LDL-c particles, and low HDL-c. The concentrations and compositions of lipoprotein particles, disturbances in functionality, and especially reverse cholesterol transport might be different among various stages of kidney impairment [24]. Thus, dyslipidemia has an important clinical significance among CKD patients with diabetes, and the significance may be influenced by kidney function, taking into account that lipid metabolism may differ according to kidney function.

In the present study, higher levels of LDL-c predicted the risk for MACEs and all-cause mortality in both early and advanced CKD patients with diabetes in the baseline models, whereas they did not predict these risks in the time-varying Cox analysis, although the results of time-varying models might be considered similar to those of previous studies that

**Table 3. Association of serum TG/HDL-c ratio with MACEs and all-cause mortality, stratified by octile categories in patients with CKD and diabetes.**

| MACE | | | Baseline model | | Time-varying model | |
|---|---|---|---|---|---|---|
| Level | N | Event | HR (95% CI) | *P* value | HR (95% CI) | *P* value |
| <1.44 | 6472 | 626 | 0.737 (0.664–0.818) | <0.001 | 0.822 (0.741–0.912) | <0.001 |
| 1.44–1.95 | 6466 | 701 | 0.758 (0.685–0.839) | <0.001 | 0.801 (0.724–0.888) | <0.001 |
| 1.95–2.45 | 6469 | 774 | 0.875 (0.793–0.965) | 0.007 | 0.911 (0.826–1.005) | 0.06 |
| 2.45–3.00 | 6488 | 825 | 0.936 (0.851–1.03) | 0.17 | 0.968 (0.879–1.065) | 0.50 |
| 3.00–3.72 | 6455 | 874 | 1 (Ref.) | | 1 (Ref.) | |
| 3.72–4.73 | 6470 | 885 | 1.025 (0.933–1.126) | 0.61 | 1.015 (0.924–1.115) | 0.75 |
| 4.73–6.57 | 6470 | 941 | 1.085 (0.989–1.191) | 0.08 | 1.03 (0.938–1.13) | 0.53 |
| ≥6.57 | 6467 | 929 | 1.106 (1.007–1.215) | 0.03 | 1.007 (0.917–1.105) | 0.88 |
| All-cause mortality | | | Baseline model | | Time-varying model | |
| <1.44 | 6472 | 899 | 0.92 (0.837,1.01) | 0.08 | 0.977 (0.889,1.073) | 0.62 |
| 1.44–1.95 | 6466 | 912 | 0.796 (0.722,0.877) | <0.001 | 0.826 (0.749,0.911) | <0.001 |
| 1.95–2.45 | 6469 | 973 | 1.02 (0.93,1.118) | 0.67 | 1.046 (0.954,1.147) | 0.33 |
| 2.45–3.00 | 6488 | 983 | 1.057 (0.964,1.158) | 0.24 | 1.075 (0.98,1.178) | 0.12 |
| 3.00–3.72 | 6455 | 903 | 1 (Ref.) | | 1 (Ref.) | |
| 3.72–4.73 | 6470 | 909 | 1.028 (0.936,1.128) | 0.56 | 1.019 (0.928,1.119) | 0.69 |
| 4.73–6.57 | 6470 | 877 | 1.013 (0.922,1.114) | 0.78 | 0.983 (0.894,1.08) | 0.72 |
| ≥6.57 | 6467 | 833 | 1.037 (0.942,1.141) | 0.46 | 0.979 (0.889,1.077) | 0.66 |

Multivariate adjustment, which adjusted for baseline characteristics of age, sex, BMI, smoking, systolic blood pressure, diastolic blood pressure, eGFR, urinary protein, serum hemoglobin and glucose level.

demonstrated either a lack of an association or an inverse association between LDL-c and both all-cause and CV mortality in patients who have CKD [22, 25–27]. The reason that this association was not observed after additional adjustment in time-varying models was probably a consequence of adjusting the effects of temporal changes in lipid levels or therapeutic agents.

Notably, a U-shaped relationship between LDL-c and all-cause mortality was observed, which was more prominent in patients with advanced CKD. The National Cholesterol Education Program (NCEP) experts classified LDL-c levels into 5 categories: <100, 100–129, 130–159, 160–189 and ≥190 mg/dL, as optimal, near-optimal, borderline high, high and very high levels, respectively, mainly based on the association between LDL-c and coronary heart disease (CHD) [19]. In the current study, however, LDL-c levels of <94 mg/dL were associated with mortality comparable to the association of the highest LDL-c category in CKD patients with diabetes. Our study suggested that the shape of association is a U-curve, especially in patients with advanced CKD, which is considered to support real-world data. This association seems to be explained in part by "reverse causality" and by the fact that lower LDL-c was, in fact, a surrogate marker of inflammation and/or malnutrition [28]. In light of these findings, the clinical significance of LDL-c is highlighted in CKD patients with diabetes, in which other comorbidities increase as the stage of CKD progresses.

We showed that there was no correlation between the serum TG/HDL-c ratio and all-cause mortality. In particular, the TG/HDL-c ratio categories did not predict the risk for either MACEs or all-cause mortality in advanced CKD patients with diabetes, except that it predicted MACE risk in early CKD patients with diabetes. This finding could be explained in part by focusing on HDL-c in the context of CKD. It is well known that high TGs and low HDL-c are risk factors for CVD in the general population, independent of LDL-c levels. Recent studies used the combination of high TGs and low HDL-C in the form of a ratio as a single marker for detecting the risk of CVD rather than using each of those individual markers alone [29, 30].

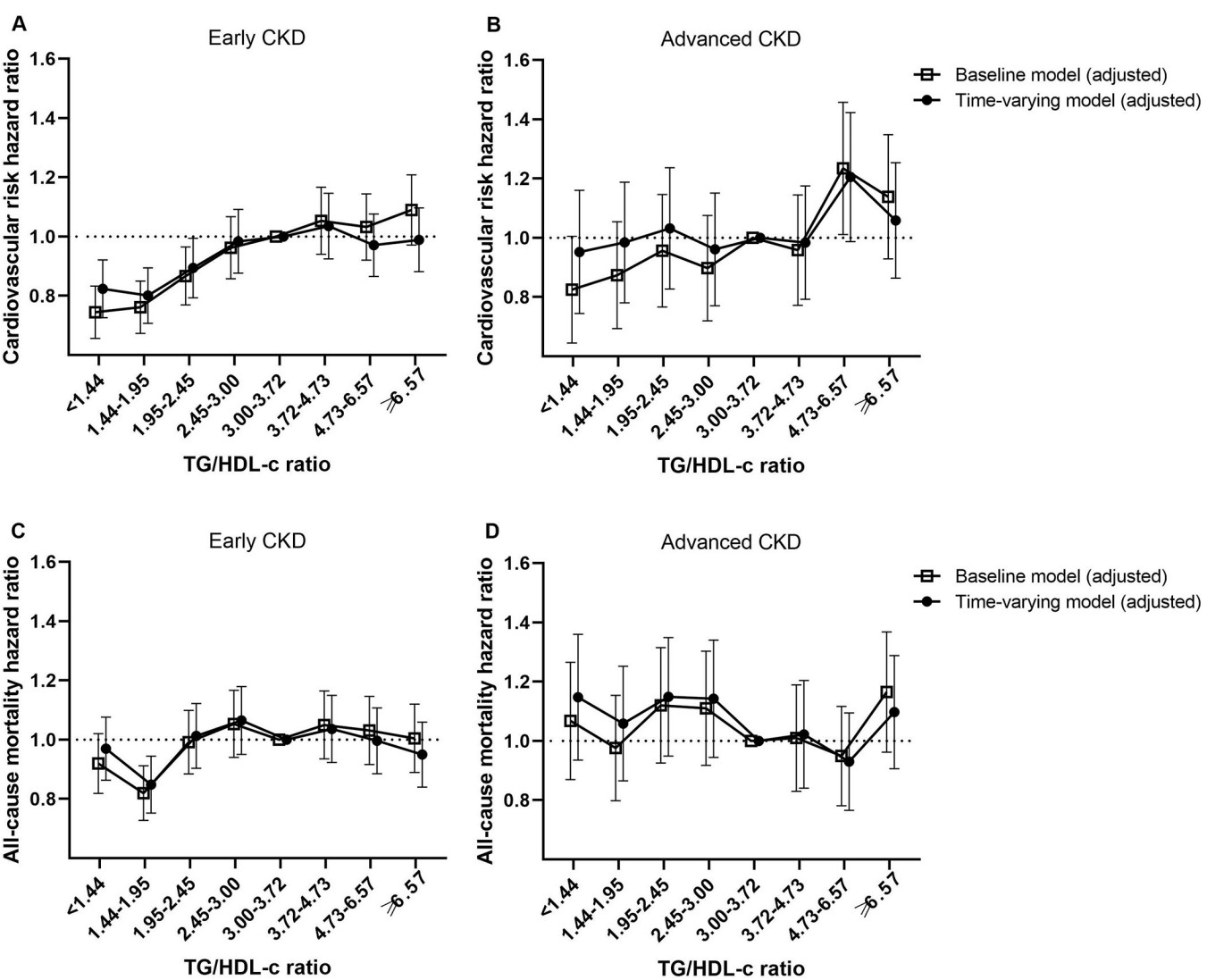

**Fig 3.** Multivariate-adjusted MACE (A and B) and all-cause mortality (C and D) hazard ratios by serum TG/HDL-c ratio in early CKD and advanced CKD patients.

However, recent studies have made it clear that increasing the plasma HDL-c concentration does not necessarily reduce cardiovascular risk. Patients with CKD tend to have alterations in both HDL-c quantity and HDL-c quality [24]. Recent studies have shown that certain conditions, such as CKD with systemic oxidative stress and inflammation, substantially reduce the capabilities of HDL-c particles and can transform them into prooxidant and proinflammatory molecules [24, 31]. This transformation has consequences for the interrelated pathways and, for example, the ability of HDL-c to prevent the oxidation of LDL-c or disturbances in reverse cholesterol transport [32].

The main strengths of this study are the large study population and the long follow-up period. The database used was stable, as it is maintained by the government or public institutions involved in providing national health information. In addition, the data included lifestyle and demographic characteristics, including smoking, alcohol consumption, physical activity, and income status, facilitating the adjustment of potential confounding factors. However, there are some limitations to the present study. First, the major limitations of the present study

originate from the inherent features of the NHIS-HIRA Cohort [33]. Second, prevalence of diseases could be underestimated since we used ICD-based diagnosis [34–36]. As the presence of diabetes was defined based on the prescription of antidiabetic medication under ICD-10 codes E11–14 or fasting glucose levels $\geq$ 126 mg, subjects with undiagnosed diabetes or those who did not visit a hospital during the study period could be omitted from these data. Also, disease severity can usually not be determined using ICD-coded data alone. It is necessary to increase the accuracy of diabetes diagnosis by also measuring HbA1c when fasting glucose is greater than 100 mg/dL in the national health screening. Last, we could not confirm diabetic nephropathy as an etiology of CKD in study patients.

In conclusion, LDL-c predicted the risk for MACE and all-cause mortality in both early and advanced CKD patients with diabetes. Very-high as well as very-low LDL-c can be a predictive marker of high mortality in patients who have CKD and diabetes, and these clinically significant findings of LDL-c were just as important in the advanced stage of CKD. However, the risk prediction of the stratified TG/HDL-c ratio was particularly inconsistent in advanced CKD. This association seems to be explained in part by a heterogeneous population with a wide range of etiologies of renal damage and by the difference in the concentrations and compositions of lipoprotein particles, disturbances in functionality, and especially reverse cholesterol transport.

## Supporting information

**S1 Table. Association of serum LDL-c with MACEs and all-cause mortality in patients with CKD and diabetes with LDL-c <79 mg/dL (1st octiles).**
(DOCX)

**S2 Table. Association of serum LDL-c with MACEs and all-cause mortality in advanced CKD patients with LDL-c <79 mg/dL (1st octiles).**
(DOCX)

**S3 Table. Association of serum LDL-c with MACE and all-cause mortality, stratified by octiles categories in CKD patients with diabetes in a statin-dropout model (for statin therapy).**
(DOCX)

**S4 Table. Association of serum LDL-c with MACE and all-cause mortality, stratified by octiles categories in early and advanced CKD with diabetes in a statin-dropout model (multivariate adjustment only).**
(DOCX)

**S5 Table. Association of serum TG/HDL-c ratio with MACE and all-cause mortality, stratified by octiles categories in CKD patients with diabetes in a statin-dropout model.**
(DOCX)

**S6 Table. Association of serum TG/HDL-c ratio with MACE and all-cause mortality, stratified by octiles categories in early and advanced CKD with diabetes in a statin-dropout model.**
(DOCX)

## Acknowledgments

This work was supported by a grant from the Korea Healthcare Technology R&D Project, the Ministry of Health and Welfare, Republic of Korea (HI17C0530). Data are available through

the Korean National Health Insurance Sharing Service. Researchers who wish to access the data can apply at (https://nhiss.nhis.or.kr/bd/ay/bdaya001iv.do) and request access to NHIS-2018-1-220.

## Author Contributions

**Conceptualization:** Yaerim Kim, Yong Chul Kim, Hajeong Lee, Yon Su Kim, Dong Ki Kim.

**Data curation:** Sehoon Park, Min Woo Kang, Semin Cho, Seoung Seok Han, Jung Pyo Lee.

**Formal analysis:** Sanghyun Park, Kyungdo Han.

**Investigation:** Soojin Lee.

**Supervision:** Kwon Wook Joo.

**Validation:** Chun Soo Lim.

**Writing – review & editing:** Yeonhee Lee, Dong Ki Kim.

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
