## [Decision Letter · Decision Letter 0]

27 Nov 2019

PONE-D-19-29456

Lipid profiles and risk of major adverse cardiovascular events in patients with CKD and diabetes: A nationwide population-based study

PLOS ONE

Dear Prof. Kim,

Thank you for submitting your manuscript to PLOS ONE. After careful consideration, we feel that it has merit but does not fully meet PLOS ONE’s publication criteria as it currently stands. Therefore, we invite you to submit a revised version of the manuscript that addresses the points raised during the review process.

We would appreciate receiving your revised manuscript by Jan 11 2020 11:59PM. To enhance the reproducibility of your results, we recommend that if applicable you deposit your laboratory protocols in protocols.io, where a protocol can be assigned its own identifier (DOI) such that it can be cited independently in the future. For instructions see: http://journals.plos.org/plosone/s/submission-guidelines#loc-laboratory-protocols

We look forward to receiving your revised manuscript.

Kind regards,

Gregory Shearer

Academic Editor

PLOS ONE

Journal Requirements:

1. In ethics statement in the manuscript and in the online submission form, please provide additional information about the patient records/samples used in your retrospective study. Specifically, please ensure that you have discussed whether all data/samples were fully anonymized before you accessed them and/or whether the IRB or ethics committee waived the requirement for informed consent. If patients provided informed written consent to have data/samples from their medical records used in research, please include this information.

2. We noticed you have some minor occurrence(s) of overlapping text with the following previous publication(s), which needs to be addressed:

https://doi.org/10.1093/eurheartj/ehy836

https://doi.org/10.1038/s41598-018-21550-3

https://doi.org/10.1093/ndt/gfv379

https://doi.org/10.1111/ceo.13157

https://doi.org/10.1038/s41598-019-46654-2.

In your revision ensure you cite all your sources (including your own works), and quote or rephrase any duplicated text outside the Methods section. Further consideration is dependent on these concerns being addressed.

Reviewers' comments:

Reviewer's Responses to Questions

**Comments to the Author**

1. Is the manuscript technically sound, and do the data support the conclusions?

Reviewer #1: Yes

Reviewer #2: Yes

2. Has the statistical analysis been performed appropriately and rigorously? 

Reviewer #1: Yes

Reviewer #2: Yes

3. Have the authors made all data underlying the findings in their manuscript fully available?

Reviewer #1: Yes

Reviewer #2: Yes

4. Is the manuscript presented in an intelligible fashion and written in standard English?

Reviewer #1: Yes

Reviewer #2: Yes

5. Review Comments to the Author

Reviewer #1: 1) title, please don't use "patients"

2)abstract, please provide numeric values (eg, OR values and 95%CIs)

3) Table 1, please remove P-value due to large sample size

4) The lowest cutoff point for LDL was 79 mg/dL. Please further look at 50 and 30 mg/dL due to the large case number

5) Please discuss the potential limitation for using ICD codes to identify cases.

Reviewer #2: Using the Korean National Health Insurance Database in this manuscript the authors have described the association of serum lipid parameters with cardiovascular outcomes in a large population of Korean )patients with mild to severe diabetic nephropathy.

They found that patients with advanced CKD (eGFR <30 mL/min/1.73 m2) had lower serum total cholesterol, LDL-c, and HDL-c but higher non-HDL-c levels and triglyceride/HDL-c ratios. They found a positive correlation between serum LDL-c and the major adverse cardiovascular events (MACE) in patients with early and advanced CKD. They further found a U-shaped relationship between serum LDL-c and all-cause mortality. The TG/HDL-c ratio showed a positive association for risk of MACE in patients with early CKD, but not advanced CKD. They found no correlation between serum TG/HDL ratio and all-cause mortality in the study population. The LDL-c level predicted the risk for MACEs and all-cause mortality in diabetic patients with early and advanced CKD, although the patterns of the association differed from each other. The TG/HDL-chol ratio did not predict the risk for either MACEs or all-cause mortality in patients with advanced diabetic nephropathy. However, the TG/HDL-c ratio predicted the risk of MACE in diabetic patients with early CKD.

The data presented on a large Korean CKD population in this paper is novel and consistent with those reported on CKD populations of other nationalities.

6. PLOS authors have the option to publish the peer review history of their article (what does this mean?). If published, this will include your full peer review and any attached files.

Reviewer #1: No

Reviewer #2: Yes: ND Vaziri

---

## [Author Response · Author response to Decision Letter 0]

10 Jan 2020

Reviewer comments: 

Reviewer #1:

1) title, please don't use "patients"

We agree with the review’s comments. As the reviewer recommended, we have removed “patients” from the title. 

2) abstract, please provide numeric values (eg, OR values and 95% CIs)

We thank the reviewer for critical comments. We have added P-for-trend for the linear association of LDL-c or TG/HDL-c ratio with HR, and we also added the lowest and highest HR (95% CI) for the U-shape relationship in the revised manuscript. 

3) Table 1, please remove P-value due to large sample size

Thank you for the reviewer’s thoughtful comments. We have removed P-values in Table 1.

4) The lowest cutoff point for LDL was 79 mg/dL. Please further look at 50 and 30 mg/dL due to the large case number

We thank the reviewer for these points. As the reviewer recommended, we have looked further at patients with LDL-c <79 mg/dL (1st octiles). Of 6,564 patients, the number of patients with the categories of LDL-c 50-78 mg/dL and 30-49 mg/dL was 5,343 and 915, respectively. Most of the 1st octile group of baseline LDL-c had LDL-c levels above 50 mg/dL, and those with extremely low LDL-c (<30 mg/dL) accounted for 0.6% of the total (306/51,757), which were very low, resulting in insufficient statistical power.

5) Please discuss the potential limitation for using ICD codes to identify cases.

We agree with the reviewer’s comment on this. The International classification of diseases (ICD) data are often used by health researchers to study health services, mortality, and other outcomes. Although validation studies can be carried out to ensure the validity of coding in administrative data, there are still limitations when using ICD information (Izeta Kurbasic et al. Acta Inform Med. 2008; 16(3): 159–161., Jetté et al. Med Care 2010;48: 1105-1110). The ICD classification is not suitable if there is few or no information about the patient. In this case, we can only code symptoms of the disease, which can be caused by several different health conditions that can be coded regularly, if we have enough information to confirm diagnosis. Consequently, ICD-based diagnosis may underestimate the true prevalence of diseases. Also, disease severity can usually not be determined using ICD-coded data alone. In a recent report, it was concluded that the ICD codes alone failed to capture many heart failure (HF) admissions, and that using the hospital codes alone would underestimate the total burden of HF (E. Ingelsson et al. The European Journal of Heart Failure (2005) 787–791). This limitation was also reported in the identification of diabetes (Khokhar B, et al. BMJ Open 2016;6:e009952). We have added this issue as a limitation in the revised manuscript. 

Reviewer #2: Using the Korean National Health Insurance Database in this manuscript the authors have described the association of serum lipid parameters with cardiovascular outcomes in a large population of Korean patients with mild to severe diabetic nephropathy.

They found that patients with advanced CKD (eGFR <30 mL/min/1.73 m2) had lower serum total cholesterol, LDL-c, and HDL-c but higher non-HDL-c levels and triglyceride/HDL-c ratios. They found a positive correlation between serum LDL-c and the major adverse cardiovascular events (MACE) in patients with early and advanced CKD. They further found a U-shaped relationship between serum LDL-c and all-cause mortality. The TG/HDL-c ratio showed a positive association for risk of MACE in patients with early CKD, but not advanced CKD. They found no correlation between serum TG/HDL ratio and all-cause mortality in the study population. The LDL-c level predicted the risk for MACEs and all-cause mortality in diabetic patients with early and advanced CKD, although the patterns of the association differed from each other. The TG/HDL-chol ratio did not predict the risk for either MACEs or all-cause mortality in patients with advanced diabetic nephropathy. However, the TG/HDL-c ratio predicted the risk of MACE in diabetic patients with early CKD.

The data presented on a large Korean CKD population in this paper is novel and consistent with those reported on CKD populations of other nationalities.

- We appreciate the reviewer for the thoughtful comment. Our study aimed to identify possible associations between lipid profile parameters and MACEs and mortality in CKD patients with diabetes and to evaluate the impact of kidney function on the associations, with large cohort and inclusiveness. Accordingly, we believe that the present data showed the consistence with those reported on CKD with diabetes populations of other nationalities.

Again, we would like to thank the reviewer very much.

---

## [Decision Letter · Decision Letter 1]

5 Feb 2020

PONE-D-19-29456R1

Lipid profiles and risk of major adverse cardiovascular events in CKD and diabetes: A nationwide population-based study

PLOS ONE

Dear Prof. Kim,

Thank you for submitting your manuscript to PLOS ONE. After careful consideration, we feel that it has merit but does not fully meet PLOS ONE’s publication criteria as it currently stands. Therefore, we invite you to submit a revised version of the manuscript that addresses the points raised during the review process.

We would appreciate receiving your revised manuscript by Mar 21 2020 11:59PM. To enhance the reproducibility of your results, we recommend that if applicable you deposit your laboratory protocols in protocols.io, where a protocol can be assigned its own identifier (DOI) such that it can be cited independently in the future. For instructions see: http://journals.plos.org/plosone/s/submission-guidelines#loc-laboratory-protocols

We look forward to receiving your revised manuscript.

Kind regards,

Gregory Shearer

Academic Editor

PLOS ONE

Additional Editor Comments (if provided):

Please be sure to carefully and completely respond to Reviewer 1's questions.

Reviewers' comments:

Reviewer's Responses to Questions

**Comments to the Author**

1. If the authors have adequately addressed your comments raised in a previous round of review and you feel that this manuscript is now acceptable for publication, you may indicate that here to bypass the “Comments to the Author” section, enter your conflict of interest statement in the “Confidential to Editor” section, and submit your "Accept" recommendation.

Reviewer #1: (No Response)

Reviewer #2: All comments have been addressed

2. Is the manuscript technically sound, and do the data support the conclusions?

Reviewer #1: Partly

Reviewer #2: Yes

3. Has the statistical analysis been performed appropriately and rigorously? 

Reviewer #1: Yes

Reviewer #2: Yes

4. Have the authors made all data underlying the findings in their manuscript fully available?

Reviewer #1: Yes

Reviewer #2: Yes

5. Is the manuscript presented in an intelligible fashion and written in standard English?

Reviewer #1: Yes

Reviewer #2: Yes

6. Review Comments to the Author

Reviewer #1: Thanks for addressing most of my concerns. However, there were still ~1200 participants with LDL <50 and 300 with LDL <30. You may have power to detect a trend or moderate-to-big difference. Statistical significance is not a major issue we should concern about. Even you did not find statistically significant results, it is still worthy to report these results as they could be practically significant.

Reviewer #2: The authors have addressed the issues I had raised satisfactorily. I therefore recommend acceptance of their paper for publication.

7. PLOS authors have the option to publish the peer review history of their article (what does this mean?). If published, this will include your full peer review and any attached files.

Reviewer #1: No

Reviewer #2: Yes: Nosratola D Vaziri MD, MACP

---

## [Author Response · Author response to Decision Letter 1]

21 Feb 2020

#1

Reviewer #1: Thanks for addressing most of my concerns. However, there were still ~1200 participants with LDL <50 and 300 with LDL <30. You may have power to detect a trend or moderate-to-big difference. Statistical significance is not a major issue we should concern about. Even you did not find statistically significant results, it is still worthy to report these results as they could be practically significant.

We thank the reviewer for these points, again. As the reviewer recommended, we have looked further at patients with LDL-c <79 mg/dL (1st octiles) and additionally analyzed subgroups with the categories of LDL-c 50-78 mg/dL, 30-49 mg/dL and <30 mg/dL. Most of the 1st octile group of baseline LDL-c had LDL-c levels above 50 mg/dL, and those with extremely low LDL-c (<30 mg/dL) accounted for 0.6% of the total (306/51,757). When compared with patients with LDL-c 50-78 mg/dL, the relative hazard of MACEs and all-cause mortality in the category of LDL-c 30-49 mg/dL was higher in both baseline and time-varying models (highest HR 1.32; 95% CI 1.13-1.55). Notably, this trend was more prominent in patients with advanced CKD (HR 1.83; 95% CI 1.36-2.45 and HR 1.80; 95% CI 1.34-2.42 in baseline and time-varying models, respectively). These additional results more firmly suggested that the shape of association between serum LDL-c and all-cause mortality is a U-curve, especially in patients with advanced CKD, which is considered to support real-world data. We have added these results in the result section of the revised manuscript.

Again, we are really grateful for your consideration.

---

## [Decision Letter · Decision Letter 2]

26 Feb 2020

PONE-D-19-29456R2

Lipid profiles and risk of major adverse cardiovascular events in CKD and diabetes: A nationwide population-based study

PLOS ONE

Dear Prof. Kim,

Thank you for submitting your manuscript to PLOS ONE. After careful consideration, we feel that it has merit but does not fully meet PLOS ONE’s publication criteria as it currently stands. Therefore, we invite you to submit a revised version of the manuscript that addresses the points raised during the review process.

We would appreciate receiving your revised manuscript by Apr 11 2020 11:59PM. To enhance the reproducibility of your results, we recommend that if applicable you deposit your laboratory protocols in protocols.io, where a protocol can be assigned its own identifier (DOI) such that it can be cited independently in the future. For instructions see: http://journals.plos.org/plosone/s/submission-guidelines#loc-laboratory-protocols

We look forward to receiving your revised manuscript.

Kind regards,

Gregory Shearer

Academic Editor

PLOS ONE

Reviewers' comments:

Reviewer's Responses to Questions

**Comments to the Author**

1. If the authors have adequately addressed your comments raised in a previous round of review and you feel that this manuscript is now acceptable for publication, you may indicate that here to bypass the “Comments to the Author” section, enter your conflict of interest statement in the “Confidential to Editor” section, and submit your "Accept" recommendation.

Reviewer #1: (No Response)

2. Is the manuscript technically sound, and do the data support the conclusions?

Reviewer #1: Partly

3. Has the statistical analysis been performed appropriately and rigorously? 

Reviewer #1: Yes

4. Have the authors made all data underlying the findings in their manuscript fully available?

Reviewer #1: Yes

5. Is the manuscript presented in an intelligible fashion and written in standard English?

Reviewer #1: Yes

6. Review Comments to the Author

Reviewer #1: Thank you for including additional analysis to look at those with low LDL-c concentration . Results are important for public health and clinical practice. The only suggestion is to add these results into your abstract. This is important because in abstract you stated that "There was a positive linear association between serum LDL-c and MACE risk in both early and advanced CKD patients (P <0.001 for trend)" (Line 42). This may not be true based on your additional analysis :" the relative hazard of MACEs and all-cause mortality in the category of LDL-c 30-49 mg/dL was higher in both baseline and time-varying models " (line 250-251).

7. PLOS authors have the option to publish the peer review history of their article (what does this mean?). If published, this will include your full peer review and any attached files.

Reviewer #1: No

---

## [Author Response · Author response to Decision Letter 2]

27 Feb 2020

Reviewer #1: Thank you for including additional analysis to look at those with low LDL-c concentration. Results are important for public health and clinical practice. The only suggestion is to add these results into your abstract. This is important because in abstract you stated that "There was a positive linear association between serum LDL-c and MACE risk in both early and advanced CKD patients (P <0.001 for trend)" (Line 42). This may not be true based on your additional analysis: "the relative hazard of MACEs and all-cause mortality in the category of LDL-c 30-49 mg/dL was higher in both baseline and time-varying models" (line 250-251).

-> Thank you for the reviewer’s thoughtful comments. We have added these results in the abstract. The sentences added in the abstract are the following. 

There was a positive linear association between serum LDL-c and MACE risk in both early and advanced CKD patients (P <0.001 for trend), except for the category of LDL-c 30-49 mg/dL in extremely low LDL-c subgroup analyses.

---

## [Decision Letter · Decision Letter 3]

23 Mar 2020

Lipid profiles and risk of major adverse cardiovascular events in CKD and diabetes: A nationwide population-based study

PONE-D-19-29456R3

Dear Dr. Kim,

We are pleased to inform you that your manuscript has been judged scientifically suitable for publication and will be formally accepted for publication once it complies with all outstanding technical requirements.

With kind regards,

Gregory Shearer

Academic Editor

PLOS ONE

Additional Editor Comments (optional):

Reviewers' comments:

Reviewer's Responses to Questions

**Comments to the Author**

1. If the authors have adequately addressed your comments raised in a previous round of review and you feel that this manuscript is now acceptable for publication, you may indicate that here to bypass the “Comments to the Author” section, enter your conflict of interest statement in the “Confidential to Editor” section, and submit your "Accept" recommendation.

Reviewer #1: All comments have been addressed

2. Is the manuscript technically sound, and do the data support the conclusions?

Reviewer #1: Yes

3. Has the statistical analysis been performed appropriately and rigorously? 

Reviewer #1: Yes

4. Have the authors made all data underlying the findings in their manuscript fully available?

Reviewer #1: Yes

5. Is the manuscript presented in an intelligible fashion and written in standard English?

Reviewer #1: Yes

6. Review Comments to the Author

Reviewer #1: The authors have adequately addressed your comments for previous version. No further comments. Suggest to accept.

7. PLOS authors have the option to publish the peer review history of their article (what does this mean?). If published, this will include your full peer review and any attached files.

Reviewer #1: No

---

## [Editor Report · Acceptance letter]

25 Mar 2020

PONE-D-19-29456R3 

Lipid profiles and risk of major adverse cardiovascular events in CKD and diabetes: A nationwide population-based study 

Dear Dr. Kim:

I am pleased to inform you that your manuscript has been deemed suitable for publication in PLOS ONE. Congratulations! Your manuscript is now with our production department. 

With kind regards,

on behalf of

Dr. Gregory Shearer 

Academic Editor

PLOS ONE